# Influence of the grounding zone on the internal structure of ice shelves

K. E. Miles [1] ✉, B. Hubbard [2], A. Luckman [3], B. Kulessa [3], S. Bevan [3], S. Thompson[4] & G. Jones [3,5]

Antarctic ice shelves typically comprise continental meteoric ice, in situ-accumulated meteoric ice, and marine ice accumulated at the shelf base. Using borehole optical televiewer logs from across Larsen C Ice Shelf, Antarctic Peninsula, we identify and report an intermediate ice unit, located between continental and in situ meteoric ice, that is tens of metres thick and formed of layers that progressively increase in dip (by ~60°) with depth. The unit's stratigraphic position and depth, supported by flowline modelling, indicate formation at the grounding zone. We hypothesise that the unit forms due to changes in the surface slope of feeder glaciers at the grounding zone, resulting in both variable surface accumulation and intense deformation. The top of the unit also marks the depth at which lateral consistency in radar layering is lost from radargrams, which may, to some degree, mark the depth of grounding zone ice across all ice shelves.

Antarctic ice shelves buttress the flow of grounded ice from the ice sheet interior. Rift propagation across ice shelves leads to mass loss through calving that, in extreme cases, can lead to ice shelf disintegration (e.g., Larsen A in 1995 and Larsen B in 2002) and accelerated inland ice discharge[1,2]. Ice shelf structure is therefore important to understand and incorporate into models projecting the longevity of ice shelves, particularly in light of Antarctic Peninsula warming and the southward progression of ice shelf disintegration[3]. While airborne and ice surface radar have been interpreted as indicating the presence of: i) basal marine ice; ii) overlying continental meteoric ice, including blue ice interpreted to have been created at the grounding zone by katabatic winds; and iii) uppermost ice shelf ice[4–8] (herein referred to as in situ meteoric ice), the technique rarely yields information on the detailed internal structure of those units. One exception is on Brunt Ice Shelf, where ground-penetrating radar surveys showed dipping layers at depth above a block interpreted as continental meteoric ice, with the dipping ice layers shallowing (by ~20°) towards the ice surface[9].

In contrast to radar, borehole observations can yield specific in situ information at higher vertical resolution. For example, borehole-based analysis on the Amery Ice Shelf revealed the thermal profile of basal marine ice[10,11], contrasting with that of the overlying meteoric ice[12,13]. On Larsen C Ice Shelf, borehole optical televiewer logs at five locations revealed the presence of in situ meteoric ice, advected continental ice (at two sites), and infiltration ice—including a massive ice unit some tens of metres thick formed by the repeated refreezing of water associated with supraglacial ponds[14,15]. Yet, despite the importance of ice shelf structure, very little is known about the internal structure of the principal units, or of the boundaries between them.

Here, we present and interpret the internal structure of two boreholes, located in the southern sector of Larsen C Ice Shelf, revealed by optical televiewer logs. The boreholes were drilled by hot water through approximately half the ice shelf thickness at each site. One borehole (120 m deep) is located in the Joerg Peninsula suture zone and named 'JP-21': an acronym of the location (Joerg Peninsula) and the approximate distance in km (21) of the site from the MODIS Mosaic of Antarctica (MOA) grounding zone position[16]. The other ('SI-47'; 160 m deep) is located ~4 km distant in Solberg Inlet meteoric ice (Fig. 1). The optical televiewer logs (which provide geometrically accurate full-colour images of the entire borehole wall) were analysed to define and characterise the principal ice units present and to

[1]Lancaster Environment Centre, Faculty of Science and Technology, Lancaster University, Lancaster, UK. [2]Centre for Glaciology, Department of Geography and Earth Sciences, Aberystwyth University, Aberystwyth, UK. [3]Department of Geography, Faculty of Science and Engineering, Swansea University, Swansea, UK. [4]Australian Antarctic Program Partnership, Institute of Marine and Antarctic Studies, University of Tasmania, Hobart, Tasmania, Australia. [5]School of Earth and Environmental Sciences, Cardiff University, Cardiff, UK. ✉e-mail: k.miles1@lancaster.ac.uk

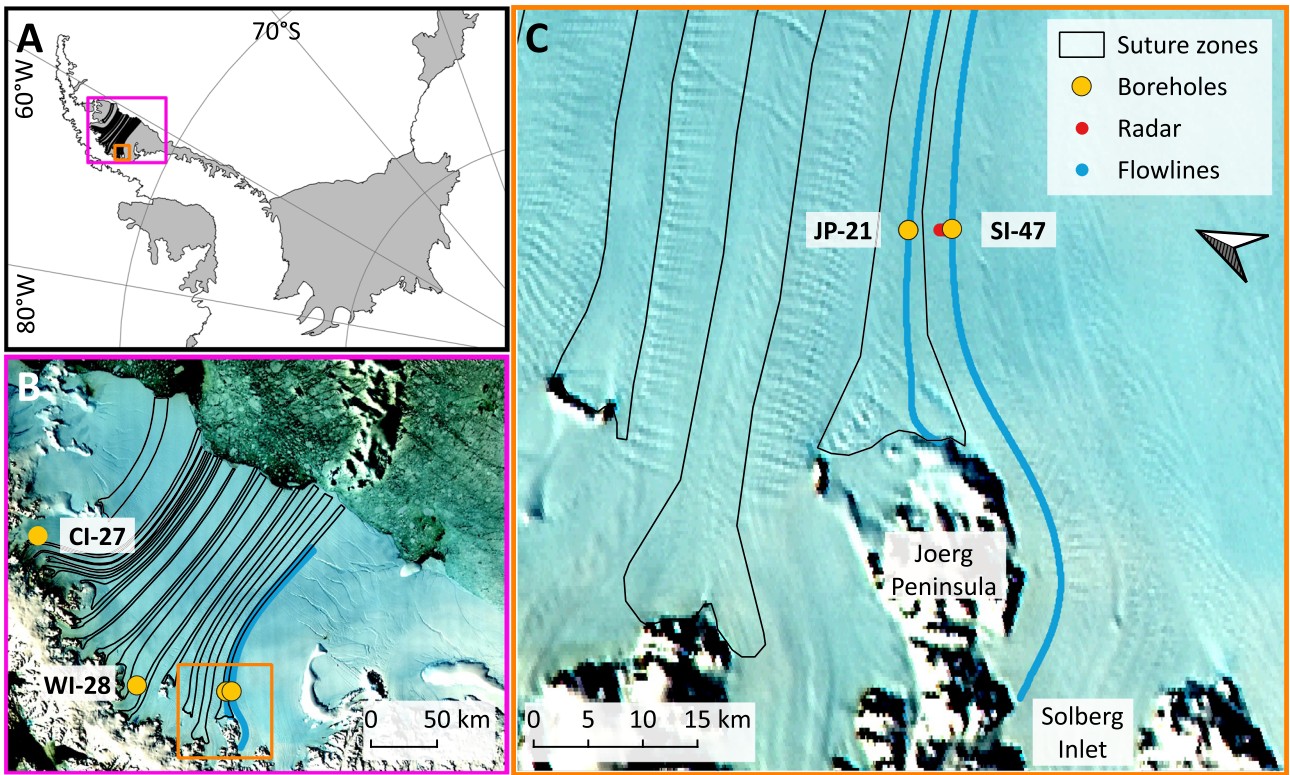

**Fig. 1 | Location of field sites. A** Location of Larsen C Ice Shelf on the Antarctic Peninsula. **B** Location of boreholes on Larsen C Ice Shelf (see text for naming convention). **C** Location of borehole sites and the ground-penetrating radar profile shown in Fig. 7. The coastline and grounding zone in **A** are from the MEaSUREs MODIS Antarctic mosaic[16] and the background of **B** and **C** are a MODIS scene from 2016. Flowlines are based on mean Sentinel-1 feature-tracked velocities for 2021 (see Methods).

identify and orientate each individual planar layer, totalling 2764, present within those units. Flowline accumulation from source to the borehole locations was also modelled to calculate the depth of ice formed at the grounding zone (see Methods). Surface ground-penetrating radar transects at 50 MHz were collected from the area around the boreholes and, finally, to extend the analysis farther, we reanalysed optical televiewer logs collected from the ice shelf's northern sector in 2015 and 2016[14,15,17]. These northern boreholes (Fig. 1B) are renamed to 'CI-27' (in Cabinet Inlet) and 'WI-28' (in Whirlwind Inlet; previously 'CI-0' and 'WI-0', respectively[14,15,17,18]), using the distances from the MOA grounding zone for consistency with the new borehole sites.

## Results

### Optical televiewer logs

Full optical televiewer logs are shown in Fig. 2 and borehole incli-nation with depth, also measured by the optical televiewer, in Supplementary fig. 1 (with neither borehole deviating > 1° from vertical beneath the water level). In the optical televiewer logs, ice layering is evident as alternating light and dark bands (indicating relatively bubble-rich and bubble-free ice, respectively)[19], typically centimetres to tens of centimetres thick. Examples of layer deli-neations are shown in Supplementary fig. 2. The upper two-thirds of the borehole logs at both JP-21 (Fig. 2A, B) and SI-47 (Fig. 2E, F) are formed almost exclusively of near-horizontal layering. Below these depths, the dip of the layers increases by over 60° over a depth range of 13 m at JP-21 and 31 m at SI-47 (Fig. 2C, D, G, H). At SI-47, the steeply dipping layers continue to the base of the borehole. At JP-21, an additional sharp transition is evident at 99 m depth where the borehole wall is mostly obscured as the light emitted from the optical televiewer is absorbed, darkening the image. Between this

transition and the base of the borehole at 120 m depth, only twelve layers are visible.

The strike and dip of all 2764 delineated layers are shown in Fig. 3, along with the following units, which were defined based on their physical characteristics. Unit 1 contains near-horizontal layers with no preferred orientation and is present from the surface to 80 m depth at JP-21 (Fig. 2B) and from the surface to 103 m depth at SI-47 (Fig. 2F). Unit 2 is defined as the immediately underlying section down which dip increases progressively and strike tends to one (at JP-21) or two (at SI-47; differing by ~180°) orientations, from 80 to 93 m depth at JP-21 (Fig. 2C) and 103 to 134 m at SI-47 (Fig. 2G). For additional visualisation, the strike and dip of Unit 2 at both sites are also shown as tadpole plots in Supplementary fig. 3. The two dominant orientations of Unit 2 at SI-47 cannot be due to instrumental error (see Methods). Unit 3 contains layers of a markedly lower dip but of the same orientation as the above Unit 2 and is only present at JP-21, from 93 to 99 m (Fig. 2D). Unit 4 is defined by layers of high dip and a single orientation, and forms the lowermost section of both boreholes, from 99 to 120 m at JP-21 and 134 to 160 m at SI-47 (Fig. 2H).The poles to planes of all delineated layers, used to carry out eigen analysis, are shown as hemispheric plots in Fig. 4. The primary eigenvalues indicate a single preferred orientation for all units (eigenvalues greater than 0.9; Supplementary Table 1) except for Unit 2 at SI-47, which has a second preferred orientation that is 177.1° different in strike from the orientation of the primary eigenvector. For this unit at SI-47, the primary eigenvalue is 0.76 and its secondary eigenvalue is 0.24. All primary eigenvectors (Supplementary Table 1) are illustrated in Fig. 5.

### Flowline accumulation modelling

Our flowline modelling predicts ice formed at the grounding zone to have been buried to a depth of 83 m (range: 45–124 m; see

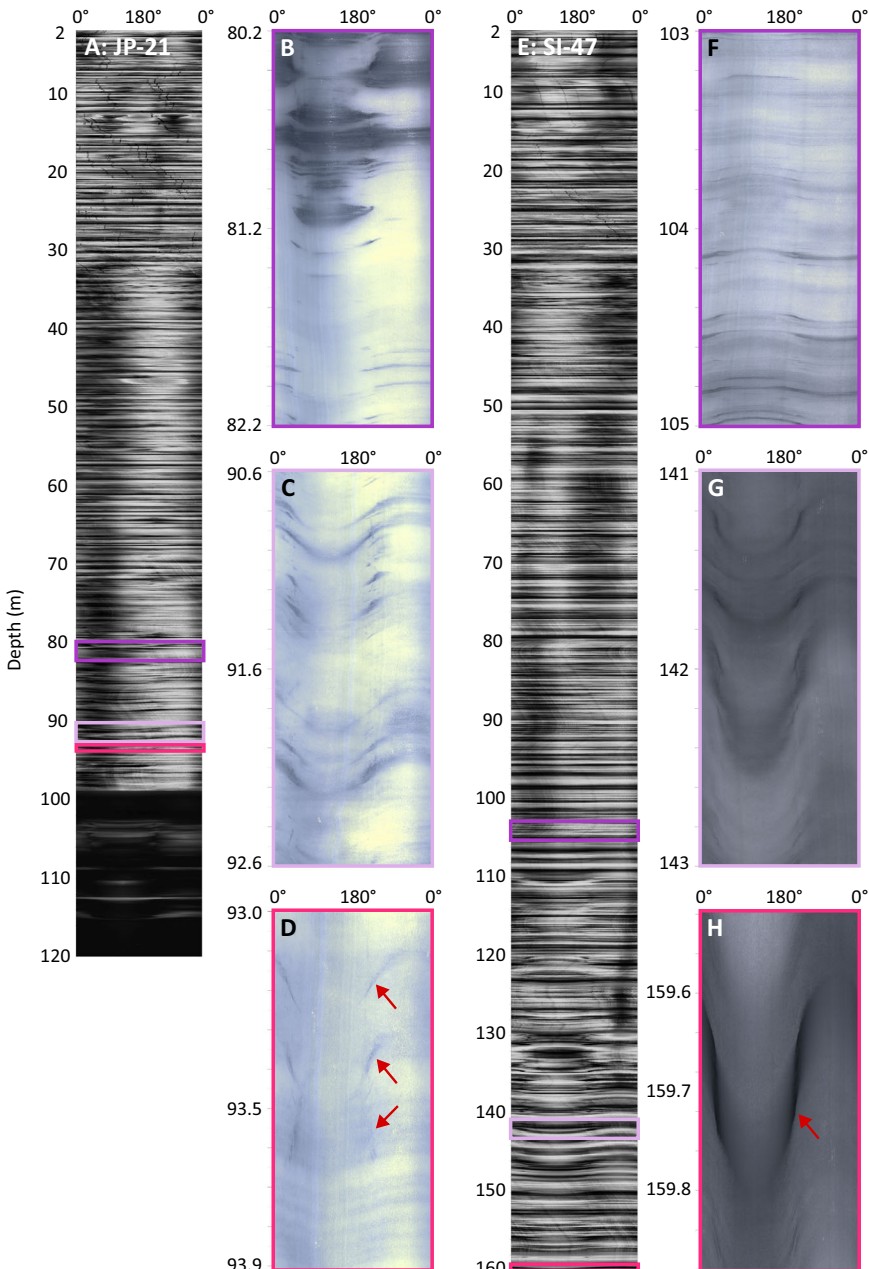

**Fig. 2 | Optical televiewer logs.** Full optical televiewer logs for JP-21 (**A**) and SI-47 (**E**). Vertical traces are superficial artefacts from the drilling process. **B** and **F** show examples of near-horizontal layering present for the top 80 m of JP-21 and 103 m of SI-47, respectively. **C** and **G** show examples of more steeply dipping layers at depth for JP-21 and SI-47, respectively. **D** and **H** show examples of crevasse traces at JP-21 (cutting across other fainter ice layers) and SI-47 (containing a bubble-free ice layer), respectively, indicated by red arrows. Layer delineations for panels **D** and **H** are shown in Supplementary fig. 2. All logs shown were recorded moving up the borehole – the downward log was indistinguishable from the upward log in each borehole – and are orientated to true North. **A** and **E** are shown in greyscale to normalise operator-controlled lighting changes made during logging; all other panels are shown in true colour at varying scales (note the differing y-axis ranges).

Methods) with an age of 139 years at JP-21, and a depth of 161 m (range: 88–237 m) with an age of 321 years at SI-47 (Fig. 6). As the surface mass balance values used are based on 1979–2014 annual means and Antarctic Peninsula snow accumulation has increased over the past few hundred years[20], it is likely that the recent values used in the analysis overestimate total accumulation and hence the depth of ice accumulated since the grounding zone. This would have a greater impact on SI-47, which is located on a longer and thus older travel path than JP-21. We also conducted this flowline modelling in reverse to locate the origin of the upper and lower boundaries of Unit 2 near the grounding zone

(Supplementary fig. 4), the results of which are consistent with the forward modelling shown in Fig. 6.

## Ground-penetrating radar

We compare our optical televiewer-derived units with a ground-penetrating radar profile that intersected borehole SI-47 (Fig. 7). Shallow-dipping internal reflecting horizons are visible from the surface to a depth that varies between ~103 m at the location of SI-47 to ~120 m farther along the profile. Unit 1 is therefore visible as near-horizontal layering through the radargram. However, there is a notable loss of consistency in radar layering from below the base of Unit 1, such

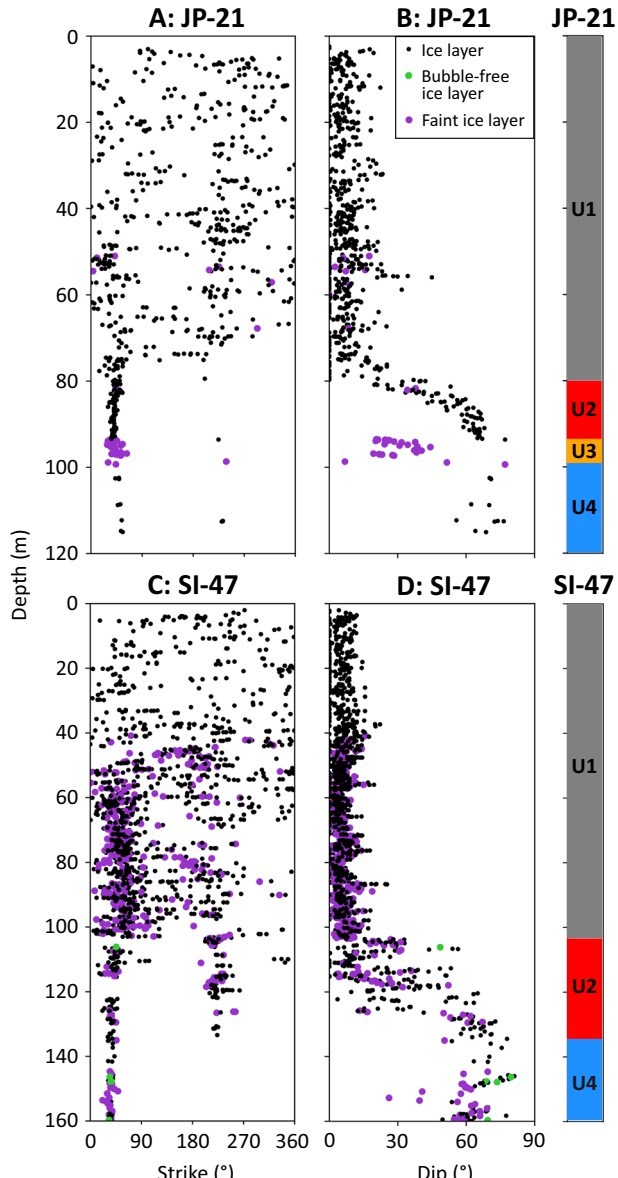

**Fig. 3 | Unit layer structure.** Strike and dip of all delineated layers from the optical televiewer logs, for JP-21 (**A** and **B**), and SI-47 (**C** and **D**), with interpreted units 1 – 4 (U1 – U4) plotted adjacent for each borehole. Examples of layer types are shown in Fig. 2. Strike is relative to true North. Source data are provided in the source data file.

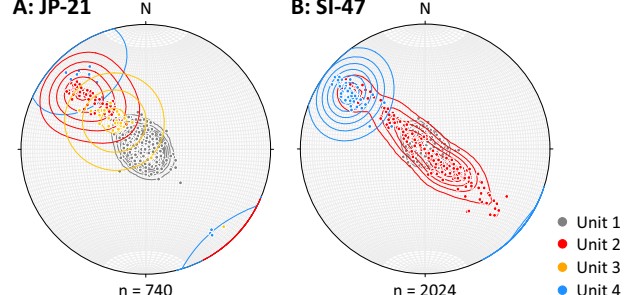

**Fig. 4 | Schmidt lower-hemisphere equal-area plots of all poles to planes.** Orientations are plotted for JP-21 (**A**) and SI-47 (**B**), coloured (following Fig. 3) and contoured (Kamb contours[34] at intervals of three standard deviations) by unit. The strike, relative to true North, is plotted as the angle, and the dip as the distance from the centre (0° at the centre and 90° at the perimeter) of the plots. Source data are provided in the source data file.

equivalent to our Unit 1. Beginning around these depths, layer dip increases progressively and develops a tendency towards one or two primary orientations, equivalent to Unit 2. At CI-27, this Unit 2 ice is < 10 m thick, but at WI-28, Unit 2 is > 40 m thick as the layers are still increasing in dip at the base of the borehole.

### Unit interpretations

Unit 1, with its regular, near-horizontal layering for the uppermost 80 m of JP-21 and 103 m of SI-47 (Fig. 3), is interpreted as in situ meteoric ice that has formed on the ice shelf surface. Most layers are typically a few centimetres thick, but a small number of almost bubble-free, thicker layers are present (e.g., the thickest layer was 0.4 m (at 28.5 m depth) at JP-21, and 1.0 m (at 47.8 m depth) at SI-47), which we interpret as infiltration ice layers. The borehole water level was observed at 35 m depth in both boreholes, similar to the pore closure depth observed in boreholes e.g., on Amery Ice Shelf[11].

Unit 4 comprises steeply dipping layers that are strongly orientated towards one primary direction at both sites (Figs. 3 and 4), and a small number of layers that: i) at JP-21, cut across the primary layer orientation (Fig. 2D); and ii) at SI-47, contain a thin and regular central band of clear (bubble-free) ice (Fig. 2H). Both i) and ii) are interpreted as crevasse traces due to their high angle and, in the case of i), the 180° strike difference between these layers and the primary layer orientation, consistent with crevasses opening orthogonal to stratification under buttressing longitudinal compression as ice crosses the grounding zone[21], and in the case of ii), visual similarities to water-healed crevasse traces interpreted elsewhere from optical televiewer logs[22,23]. On the basis of the stratigraphic position and depth of this unit, as well as the presence of crevasse traces, we interpret it as continental meteoric ice from the grounded glaciers that feed this section of the ice shelf (Fig. 1). The steep dip of the layers in the unit were likely inherited from up-flow (Supplementary fig. 5), supplemented by glacier flow over the steep bedrock approaching the grounding zone.

Unit 2 is located between overlying Unit 1 (in situ meteoric ice, accumulated downflow of the grounding zone) and underlying Unit 4 (continental meteoric ice, accumulated upflow of the grounding zone) and, as such, must have formed between the two, at or close to the grounding zone. Unit 2 comprises ice layers that dip increasingly with depth such that the unit's boundaries are conformable with both adjacent units (i.e., being sub-horizontal at the upper boundary and steep at the lower boundary). These dips tend to one dominant orientation at JP-21 and two dominant orientations (~180° different in strike) at SI-47 (Figs. 3 and 4, Supplementary fig. 3). Although subject to uncertainty in reconstructed velocity and accumulation fields, flowline modelling confirms that the depth of Unit 2 is consistent with its

that, while individual horizons can be demarcated clearly through Unit 1, they cannot be identified through Units 2–4.

### Reanalysed optical televiewer logs from northern Larsen C Ice Shelf

To investigate the spatial continuity and properties of the material units reported above elsewhere within Larsen C Ice Shelf, we analysed layers present within optical televiewer logs acquired previously from the northern sector of the ice shelf, at CI-27 and WI-28 (Fig. 1B)[14,15,17]. In general, fewer primary layers were identified in these logs, due to the greater influence of melt and infiltration ice layers that disturb the original ice layering in this sector of the ice shelf. Yet, despite this additional complexity, a similar pattern was present (Fig. 8) to those at the JP-21 and SI-47 boreholes. In CI-27 and WI-28, near-horizontal in situ meteoric ice layers were present to 45 m and 65 m depth, respectively, interpreted in the original analysis as the depths at which the meteoric ice below was transported over the grounding zone[15] – and hence

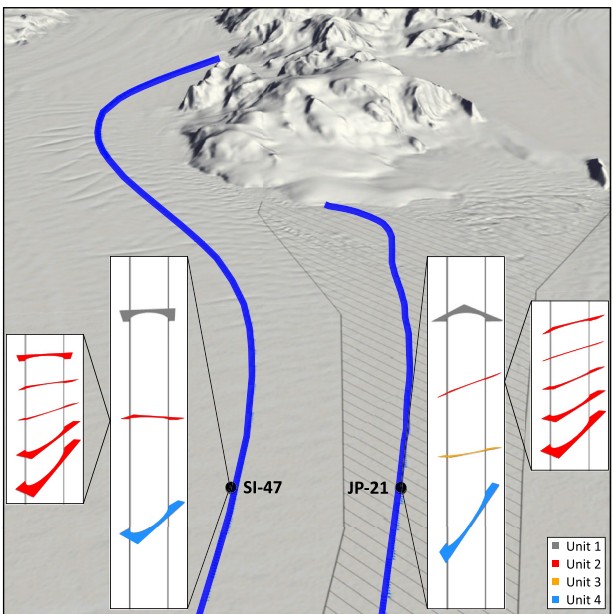

**Fig. 5 | 3D view of Larsen C Ice Shelf and Joerg Peninsula and the predominant orientation of each unit at each site.** The main insets show the primary eigenvector for each unit (Supplementary Table 1), coloured as in Fig. 3. As layers dip progressively through Unit 2, the unit is expanded further, with five layers selected at equal intervals to illustrate the progressive increase in dip with depth down the unit. While most Unit 2 layers conform to the orientations shown in the expanded panel for SI-47, several are oriented at ~180° to this orientation while maintaining a similarly increasing dip with depth (see Fig. 3C and D and tadpole plot in Supplementary fig. 3). Dark blue lines indicate the flowline for each site, produced from mean Sentinel-1 feature-tracked velocities for 2021. The view direction is from Antarctic Polar Stereographic North to South (geographic East to West). The hillshade of surface elevation is from TanDEM-X interferometry using scenes from 2012 and 2022 (exaggerated vertically by a factor of 3).

formation at or near the grounding zone. The depth of ice formed at the grounding zone is modelled to be 83 m (range: 45–124 m) for JP-21 and 161 m (range: 88–237 m) depth at SI-47 (Fig. 6), similar to the depth of the base of Unit 2 at 93 m at JP-21 and 134 m at SI-47 (considering the likely overestimation of modelled depth, particularly for SI-47).

Any process advanced for the formation of Unit 2 needs to explain as many of its distinctive properties as possible. These properties are that it is: i) located at the top of the ice column at the grounding zone; ii) tens of metres thick; iii) comprised of layers that shallow, from ~60° to ~0°, upwards, and are hence conformable with Unit 1 layering above and Unit 4 layering below; and iv) comprised of layers that either have a single, dominant orientation or have two dominant orientations that are ~180° different. While we know of no documented processes that can explain all of these properties, we hypothesise that Unit 2 forms through a combination of accumulation across a shallowing surface slope and intense deformation, both resulting from the substantial change in slope as ice flows from its steep grounded feeder glaciers into the approximately horizontal ice shelf (Supplementary fig. 5). Key elements with the capacity to create variable accumulation across a shallowing surface are outlined schematically in Fig. 9. Here, in the grounding zone, Unit 4 is tilted to dip steeply while flowing over the steep bedrock below, and thus the first layers to accumulate on top of this unit maintain a similar dip and strike. Over time and with distance over the grounding zone, the surface slope of the ice shelf shallows, and subsequent accumulating layers also decrease in dip until the surface is near-horizontal. Such a process would result in a generally progressive increase in dip with depth – characteristic of Unit 2 at all sites where it has been logged by optical televiewer (i.e., JP-21 and SI-47

in the south and CI-27 and WI-28 in the north of Larsen C Ice Shelf; Fig. 1). However, this process alone does not explain the significant minority of layers in Unit 2 at SI-47 with a strike rotated 180° from that of the dominant layering. These secondary layers are challenging to explain, particularly since they appear at SI-47 and not at JP-21. In the absence of additional evidence, we propose tentatively that these secondary layers form as fractures opening orthogonal to the main layering at locations where longitudinal compression is sufficiently intense to buckle and fracture the surface layers of ice as it flows into the shelf. In general, ice shelf strain is high across the grounding zone[4,7,24,25] and particularly high along the edges of promontories where steep and fast-moving subsidiary glaciers emerge into the main trunk (Supplementary fig. 6). Formation of Unit 2 in such a setting is consistent with the reconstructed flowline of the ice intercepted by SI-47 (Fig. 1C), which indicates formation along the lateral margin of Solberg Inlet. In contrast, the ice intercepted by JP-21 crosses the grounding zone at the tip of Joerg Peninsula, a zone of more subdued longitudinal compression.

Finally, at JP-21, an additional distinctive unit forms a layer, ~6 m thick, located below Unit 2. This Unit 3 is characterised by layers of the same orientation but lower dip than Unit 4 that were fainter in appearance than many of the ice layers elsewhere in the image logs. We interpret a small number of layers that dip more steeply and/or cut across the primary layer orientation in this unit (Figs. 2D, 3, 4) as crevasse traces, similar to those present within Unit 4, indicating an origin for Unit 3 as continental meteoric ice. The presence of Unit 3 only in the suture zone of JP-21, and not the meteoric ice band of SI-47, suggests an intricacy particular to this part of the suture zone – such as a block of continental meteoric ice that broke away from the grounded ice and was tilted as it joined the floating part of the ice shelf. The crevasse traces could be a remnant from when the ice was grounded or have formed once the block was transported over the grounding zone, within the zone of transverse crevasses associated with flexure in the floating shelf.

## Discussion

Our interpretation that Unit 2 forms during ice transport across the grounding zone (Fig. 9) implies that such a unit, forming the transition between continental meteoric ice and in situ meteoric ice, should be present at all ice shelves exhibiting steep bedrock topography at the grounding zone. Indeed, a similar structure is apparent in ground-penetrating radar surveys on Brunt Ice Shelf: dipping layers at depth, above a block interpreted as continental meteoric ice (akin to our Unit 4), shallowing towards the ice surface (akin to our Units 1 and 2)[9]. The exact thickness of Unit 2 will vary between flow units and inlet grounding zones, even on the same ice shelf (as shown by our four optical televiewer logs across Larsen C Ice Shelf), and will depend on various factors, including accumulation rate and ice velocity across the grounding zone, the geometry of the grounding zone, and subsequent vertical strain along the ice shelf flowline.

Notably, radar reflections from internal horizons lose their lateral continuity within Unit 2 (Fig. 7). This loss of radar energy and/or continuity at depth also occurs elsewhere on Larsen C[5] and at other Antarctic ice shelves, such as Brunt Ice Shelf[9] where no structure can be interpreted within continental meteoric ice blocks (e.g., their[9] Fig. 5). We observe that the depth of this loss of radar layer continuity occurs at the top of Unit 2, where ice layers begin to dip steeply (> ~20° based on layers in borehole SI-47). It is therefore possible that the Unit 2 transition from the sub-horizontal layering of Unit 1 to the steeply dipping layering of Unit 4 contributes to a notable reduction in the lateral continuity of returned radar energy, at least under the standard bistatic perpendicular-broadside antenna configuration used herein (see Methods). Yet, it is also possible that other factors influenced by the unit transition – including a change in anisotropy, destructive interference in trace stacking, and off-nadir ray path losses – may also

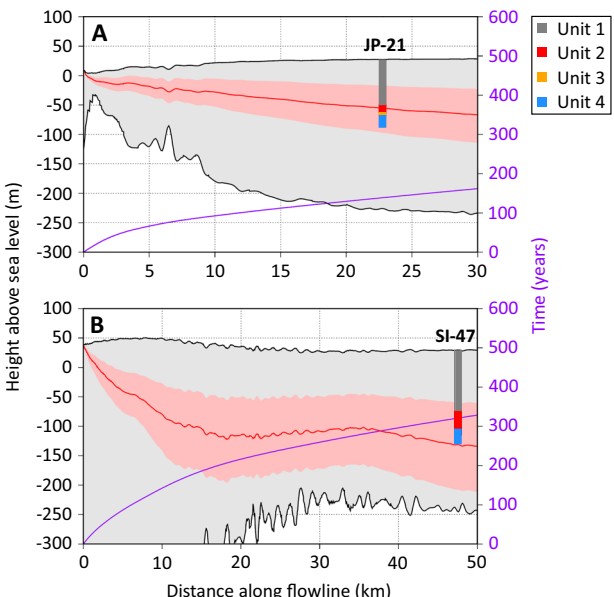

**Fig. 6 | Flowline model outputs.** Plots illustrate grounding zone depth and age profile at the borehole sites for **A** JP-21, and **B** SI-47. Boreholes are coloured by unit (following Fig. 3), with the grounding zone estimated from the borehole data to be at the base of Unit 2. The median model estimate of the grounding zone depth is shown by the red line, with the range between the maximum and minimum esti- mates (see Methods) shown shaded in light red. Time in years since ice crossed the grounding zone is shown by the secondary purple axis and line. Upper and lower ice surfaces are taken from MEaSUREs Bedmachine Antarctica V.3[35]. Source data are provided in the source data file.

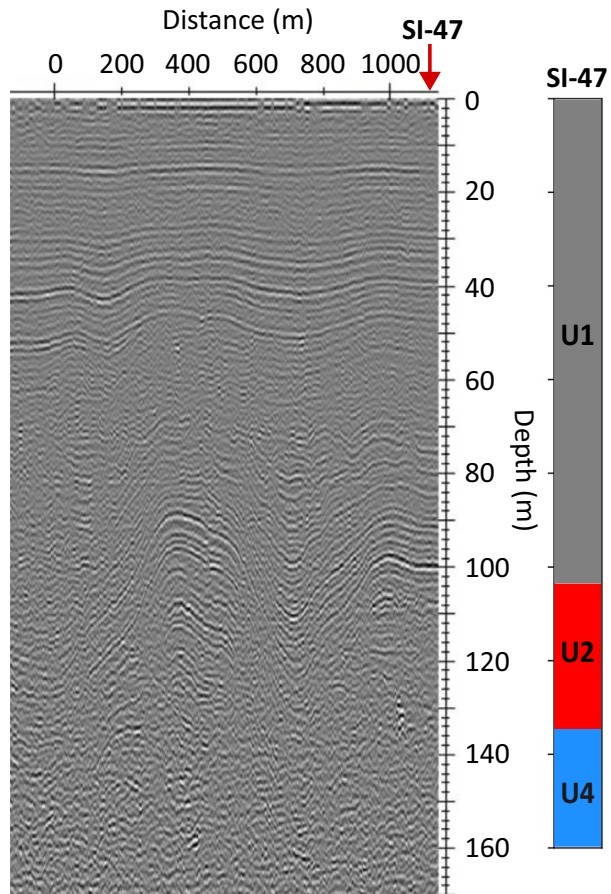

**Fig. 7 | Flow-transverse radargram intersecting borehole SI-47.** The location of SI-47 is indicated by the vertical red arrow, with interpreted borehole units (Fig. 3) plotted adjacent. Radargram location is shown in Fig. 1C.

increase the heterogeneity of radar energy return through Unit 2[26]. Regardless of the precise cause(s), our observations have implications for the interpretation of radargrams and for the potential of using radar to observe ice structural properties at depth within ice shelves. It also means that the depth of loss of radar energy and/or continuity may be used as an internal marker across ice shelves for ice formed at the grounding zone.

The suggested formation mechanism for Unit 3 (as a block of continental meteoric ice that broke away and tilted as it became buoyant) provides an additional means by which seawater could infiltrate into ice shelves. Indeed, a video log of JP-21 (acquired immediately after logging by optical televiewer) shows a dense layer of platelet ice in the borehole water at 99 m depth, which continues with varying densities of platelets to the base of the borehole at 120 m (Supplementary fig. 7). The platelets would absorb the light emitted by the optical televiewer and result in the darkening of the image seen at JP-21 from 99 m depth (Fig. 2A), with the borehole wall (i.e., Unit 4) only viewed beneath this depth where the density of platelets decreases. Where the wall is observed, the ice is of continental meteoric origin (Unit 4), and we therefore suggest that the platelets arise from the buoyant freezing of infiltrated brine as the ice is partly and temporarily submerged beneath sea level while it flows over the grounding zone in the suture zone. If Unit 3 is a separate, tilted block of continental meteoric ice, this platelet ice layer would form as its lower boundary – i.e., 99 m depth, the upper surface of Unit 4.

We note that our proposed unit formation interpretations leave certain unresolved issues. First, why Unit 2 layer orientation is similar at both JP-21 and SI-47, despite their contrasting flowline directions at the grounding zone (Fig. 5). In the absence of further evidence, we consider this to be a geometrical coincidence. Second, why the orientation of Units 2 – 4 continues to exert an influence on the orientation of Unit 1, even once layers are accumulating almost hor- izontally. This preferred orientation does weaken progressively

upwards through Unit 1, so it is possible that the dip inherited by Unit 2 from units below is also inherited by the lower layers of Unit 1, but eventually this influences wanes. This upwards influence could be related to the surface rumples formed as ice flows over the grounding zone and out onto the ice shelf (Supplementary fig. 6). Such surface rumples are initially steep but gradually reduce as subsequent snow accumulates during flow away from the grounding zone. Finally, our proposed formation mechanism for Unit 2 (Fig. 9) does not account for rebound or flexure as the ice becomes buoyant, which would act to decrease the dip of the layers in Units 2 – 4. However, we estimate that this will not have a large effect, particularly if the dip of Unit 4 layers is high prior to becoming buoyant – though the internal structure of the feeder glaciers is unknown.

Our detection of Unit 2 in four boreholes across Larsen C Ice Shelf (and identification of a similar unit from radar surveys on Brunt Ice Shelf) and interpretation of formation during transport over the grounding zone suggests that the unit is likely to be present in all ice shelves. This has important implications, as it would allow the depth of in situ meteoric ice to be determined: i) from the dip of planar ice layers, e.g., in borehole logs; and ii) from the loss of radar energy and/ or lateral continuity in ground-penetrating radar surveys. However, this loss of radar clarity also precludes radar-based structural analysis of the unit. We therefore recommend that future ground-penetrating radar surveys on ice shelves should consider alternative modes of data acquisition, such as dipole antennas mounted in a bistatic parallel- broadside configuration[27] or wide-angle radar reflection[28]. Our optical televiewer borehole logs indicate that the boundary between con- tinental and in situ meteoric ice is not sharp, but that the transition

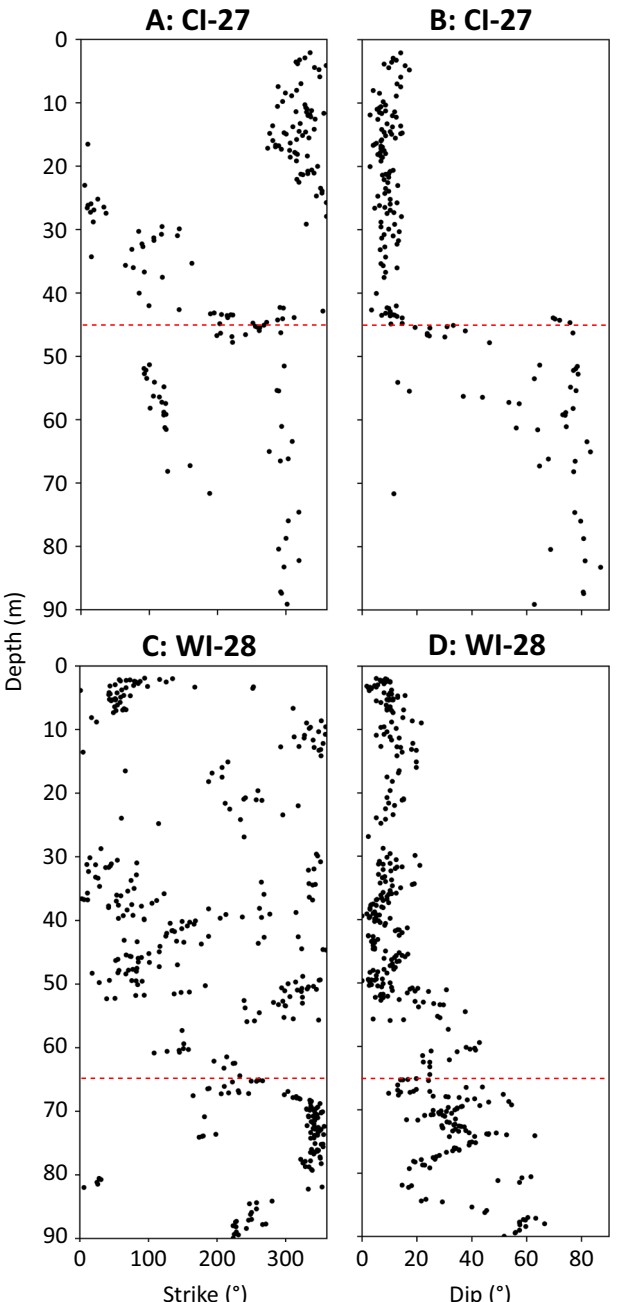

**Fig. 8 | Unit layer structure from the northern Larsen C boreholes.** Strike and dip of all delineated layers from the northern Larsen C optical televiewer logs, for CI-27 (**A** and **B**), and WI-28 (**C** and **D**). The red dashed lines indicate where the original analysis interpreted the transition between in situ (ice shelf-derived) meteoric ice (akin to our Unit 1) and continental meteoric ice derived from the grounded feeder glaciers (akin to our Unit 4), i.e., the depth of ice first formed at the grounding zone[15]. Strike is relative to true North. Source data are provided in the source data file.

between these units occurs progressively over some metres to tens of metres, with implications for ice shelf mechanics, the quantification of which awaits the unit's sampling and physical analysis.

## Methods

Fieldwork was carried out in November and December 2022 on Larsen C Ice Shelf, Antarctic Peninsula. Boreholes were hot-water drilled at two sites: JP-21 in a suture zone ~21 km downflow from the Joerg Peninsula; and SI-47 in the meteoric ice to the south of the Joerg Peninsula suture zone (Fig. 1). Once drilled, boreholes were logged by

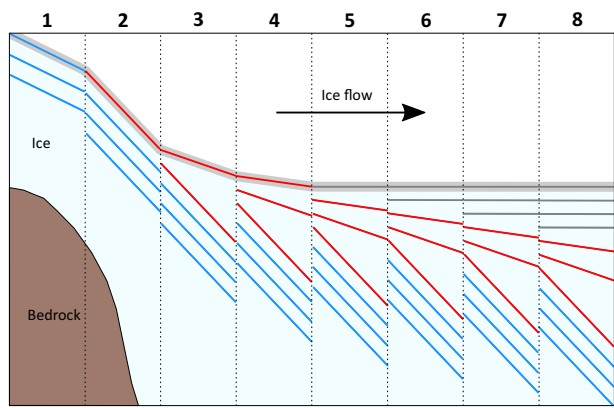

**Fig. 9 | Conceptual illustration of the proposed mechanism to create variable accumulation across a shallowing surface around the grounding zone of an ice shelf.** Ice flows from left to right, accumulating as it flows along a concave-upwards surface trajectory (thick grey line) over eight time periods. During #1, Unit 4 accumulates (blue lines). During #2, Unit 4 rotates to dip steeply as it flows down the steep-gradient bedrock over the grounding zone, and Unit 2 (red lines) starts to accumulate at this steep angle. As the surface slope shallows through #3 and #4, Unit 2 becomes increasingly less steeply dipping as it accumulates. During #5 to #8, the ice shelf surface is approximately horizontal and accumulating layers of Unit 1 (grey lines) also remain approximately horizontal. Unit 3 in the JP-21 borehole would be located between Units 4 (blue) and 2 (red) but is not included here as it is not present at both sites.

an optical televiewer to 120 m at JP-21 and 160 m at SI-47 in both down and up directions at a horizontal and vertical image resolution of ~1 mm per pixel. Image logs are automatically orientated to magnetic North by the optical televiewer's on-board magnetometers and were corrected to true North before beginning analysis. For reference, the inclination of each borehole, measured by the optical televiewer during logging, is shown in Supplementary fig. 1. Planar ice layers, present in the raw image logs as sinusoids[19], were delineated in WellCAD software to obtain each layer's strike (reported using the right-hand rule) and dip (example layer delineations are shown in Supplementary fig. 2). Four units were defined across the two sites based on the physical characteristics of the layers, and eigen analysis calculated on the poles to planes of each unit using Stereonet software[29]. Layer properties (orientation and strike) were consistent between the downward- and upward-direction logs in each borehole, excluding instrumental error as a possible cause for the two dominant orientations of Unit 2 at SI-47. The same layer delineation and analysis was carried out on the previously acquired and reanalysed optical televiewer logs from northern Larsen C Ice Shelf[14,15,17].

Flowline modelling was carried out to confirm the depth of ice accumulated since travelling over the grounding zone at each of our borehole sites, using the model in Bevan et al.[18] adapted to our borehole flowlines. The flowlines were based on mean Sentinel-1 feature-tracked velocities for 2021, comprising the average of 122 six-day and twelve-day repeat velocity maps. There is no evidence of flow rearrangement on Larsen C Ice Shelf[25], so it is reasonable to assume that this velocity map is valid for history of ice investigated here. Along each flowline, surface mass balance was accumulated using the median pixel value (and minimum and maximum for uncertainty ranges) of RACMO2 1979 – 2014 annual means (5.5 km resolution for the Antarctic Peninsula)[30]. Every 100 m along each flowline, a new accumulated ice thickness was calculated according to equation 5 from McGrath et al.[6], which considers the upstream thickness, strain thinning rate, and accumulation rate. Horizontal advection and strain thinning rates were based on Sentinel-1 velocities. Accumulated thicknesses were adjusted so that the mean density matches that from a borehole in the centre of Larsen C Ice Shelf[15]. We also conducted this flowline modelling in reverse to

locate the origin of the upper and lower boundaries of Unit 2 near the grounding zone – the uncertainty ranges for SI-47 are the same as above, but for JP-21 represent the 10–90[th] percentiles of surface mass balance values for the upper boundary of Unit 2 and 25–75[th] percentiles of surface mass balance values for the lower boundary of Unit 2, as the flowline modelling for JP-21 could not start high enough that the same uncertainty bounds could be used.

The ground-penetrating radar data were acquired around the borehole sites using tried and tested instrumentation and procedures[4,5]. A Sensors & Software PulseEkko Pro system with a high-power transmitter (transmitter output: 1000 V) and 50 MHz antennas was towed at an average speed of ~12 km h$^{-1}$ using a snowmobile and sledge assembly. In the bistatic configuration, the dipole antennas were mounted in the common perpendicular-broadside configuration[31] on a plastic sledge, the dimensions of which limited the transmitter and receiver antenna separation to 1.4 m. Sampling interval was 1.2 ns and each trace represents a distance-average stack of eight individual traces, resulting in an average trace spacing of ~2.5 m. Precise (± 0.1 m) planimetric location of each trace was achieved with a Leica VIVA GS10 Professional GNSS receiver, mounted on the snowmobile. The raw radar data were processed using standard techniques implemented in the commercial Reflex-W package, including frequency filters, correction for energy decay, background removal, Stolt (f-k) migration, and 2-D mean filtering. Travel times were converted to depth assuming a depth-averaged radar velocity of 0.173 m ns$^{-1}$, consistent with previous estimates from common-midpoint data[4,5].

## Data availability

Data are available on the NERC Polar Data Centre: optical televiewer logs[32] at https://doi.org/10.5285/957fb0bd-6d51-46fc-b4fa-731d21731eda; ground-penetrating radar data[33] at https://doi.org/10.5285/9c46ec89-e2da-4140-a8a5-443911fe34cc; and optical televiewer logs[17] from northern Larsen C Ice Shelf collected in 2014 – 2015 at https://doi.org/10.5285/5f545c54-6d76-4328-ba10-9a358456f035. TanDEM-X data were supplied by DLR. Sentinel-1 SAR data are available through the Copernicus Open Data Hub (https://dataspace.copernicus.eu/browser/). Source data are provided with this paper.

## Code availability

The code for the flowline model is available at https://doi.org/10.5194/tc-11-2743-2017.

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

## Acknowledgements

This research was funded by the "Rift Propagation for Ice sheet models (RiPlce)" Natural Environment Research Council grant awarded to Aberystwyth and Swansea Universities (NE/T008016/1). We thank the British Antarctic Survey for logistical support, particularly the project's Field Guides: Samuel Hunt, Sarah Crowsley, and Min Willis.

## Author contributions

A.L. led NERC project NE/T008016/1, for which B.H. and B.K. were co-investigators and K.M., G.J., and S.B. were post-doctoral research associates. Fieldwork was carried out by K.M., B.H., A.L., B.K., and S.T.; B.H. led the borehole drilling. K.M. and B.H. obtained, analysed, and interpreted the optical televiewer imagery. S.B. conducted the flowline accumulation modelling. K.M. and B.K. processed and analysed the ground-penetrating radar data. K.M., B.H., A.L., B.K., S.B., S.T., and G.J. contributed to data interpretations and manuscript writing, which were led by K.M.

## Competing interests

The authors declare no competing interests.
