## [Transparent Peer Review file · Nature Communications]

Influence of the grounding zone on the internal structure of ice shelves

Corresponding Author: Dr Katie Miles

Version 0:

Reviewer comments:

Reviewer #1

(Remarks to the Author)
Please see attached report.

Reviewer #2

(Remarks to the Author)
I co-reviewed this manuscript with one of the reviewers who provided the listed reports. This is part of the Nature Communications initiative to facilitate training in peer review and to provide appropriate recognition for Early Career Researchers who co-review manuscripts.

Reviewer #3

(Remarks to the Author)
I co-reviewed this manuscript with one of the reviewers who provided the listed reports. This is part of the Nature Communications initiative to facilitate training in peer review and to provide appropriate recognition for Early Career Researchers who co-review manuscripts.

Reviewer #4

(Remarks to the Author)
Report on Miles et al: "Influence of grounding zone processes on the internal structure of ice shelves"

In this manuscript borehole televiewer logs from the Larsen C ice shelf are evaluated, from hot water drills close to the grounding zone in the southern part of the ice shelf. The data reveal a change in the dip of internal layers, which is then interpreted as an inherited feature from the transition at the grounding zone.

I think this could be an interesting study, but I have some major concerns, which I hope can be addressed by the authors.

1. I do not understand the definition of the Units. This definition seems to be based on Figure 3. I tried to understand the apparent two-mode status (with a 180 degree difference, which made me suspicious) of the strike direction. This became then clear by looking at Figure 4, the stereoplots for the planes of the layers. From here it is clear that the poles to planes of Unit 2 of SI-47 are undulating around zero, which explains the change in strike by 180 degrees. I think the stereoplots give a much better illustration of the geometry, and are also more suitable to define Units. Here it looks like the dip of planes of the layers in SI-47 are undulating back and forth, until finally stagnating towards one side with a high dip angle. For JP-21 this stage is missing, and the dip is just increasing with depth in the same direction (except for the funny Unit 3). I think this should be taken into account when interpreting the data, and it might change conclusions, as in that case there would be no gradual change of dip in one direction, as shown in fig. 9. What I did not get from the manuscript: Is the strike direction relative to geographic north? How does this then relate to the flow direction?

2. I think there is some essential information missing. Is the bore hole shape logged? There is no information on how the borehole diameter changes, how the inclination of the borehole changes. This might well have an impact on the appearance

of the layers in the images.

3. Is the image log somehow orientated? Can the logger rotate in the hole, or is this somehow corrected?

4. The interpretation that the inclined layers are linked to the slope of the glaciers before entering the ice shelf is interesting, but I think it would be good to have a look at the actual slopes and show some data here. 60 degrees seems quite a lot. Why is the "final" dip angle so similar in both boreholes, when one is from a rotated block in the suture zone, and one from a smooth transition?

5. I think in the transition from inclined to horizontal layers in the transition zone (shown in figure 9) should be discussed a little more. Is this only geometry, or locally variable accumulation (why should it be different?)

comments on the figures:

- in figure 3, 4, 6 and 8, it would be really nice to have the names of the boreholes in the panels, instead just small in the figure caption.
- In figure three it is difficult to see the few colored dots in the more crowded parts. Maybe make them also bigger to stand out?
- For figure 2 it would be good to add a simple sketch of the geometry, for example the cylindrical borehole with an oblique plane intersecting it, and then "unroll" it to show the resulting curve. It would be also good to show in the zoom-panels which curves are picked from these. For me it is also not that obvious what is crevasse and what is not in panel D and H, so maybe annotate or add an arrow. And maybe stick to the same depth scale as the other panels.
- When looking at the lower part of the logs in figure 2 the dip-angles should be similar (as shown in fig 3), but JP-21 looks considerably less wavy (above the black part of course). Is this just an illusion? Maybe it would be good to add a supplementary figure with a comparison, with a scale between the overview image and the zoom panels.
- Figure 1: The background optical image does not show a lot of structure of the ice shelf surface. Maybe it would be better to use a radar image, where the flow units and suture zone can be better distinguished.

Version 1:

Reviewer comments:

Reviewer #1

(Remarks to the Author)

After the revision, the authors address most of the comments of the original submission. However, some of the additions were not clearly indicated in their response letter. Also, we think that some of the responses could be included in the final version, for in case other readers have the same questions.

Regarding the major comments:

All were nicely addressed, except the next two:

1. Modelling and evidence for the mechanism...

Figures 1 and 2 of the response could be included as supplementary figures. They are also very illustrative and worth including if the authors do not want to replace the original document, and would support the authors' suggested location of formation, as we outlined in our initial review. If the authors want to include them, are the percentiles 10th and 25th relative to the maximum or to the minimum accumulation rates?

In the response, authors say 'The analysis, whether run forward to the borehole or backwards from it, indicates clearly that it is consistent. In reality, the analysis is indeed subject to substantial uncertainty', and we think this is important to be mentioned in the article. Maybe could it be included in the paragraph between lines 180-189 (where the additions on the formation of Unit 2)?

The new lines 321-323 for further detailing the survey greatly help the reader. We would rephrase with 'In the bistatic configuration, the dipole antennas were mounted... ', or similar, as the antennas are not bistatic, but the system.

2. Radar coherency

The response and the revised text address our concerns, except when we asked about the possibility of determining the strike and dip from the radargram. In lines 140-141 and 234-249 the authors say that Units 2 and 4 cannot be identified in the radargram of Figure 7, but maybe within Unit 1 can the strike and dip be estimated?

Regarding the minor comments:

All were well addressed, except the next two:

1. The sub-figures could guide...

Except in Figure 8, we do not see some of the changes we recommended, but if the authors think this way is better, we accept it.

2. Line 287: Instead of V, would it be W (watts)?

We might be misinterpreting what the authors say, but power units are watts, not volts as seems from V units. Although this is not related to the quality of the article, we think it should be corrected.

A few more minor comments which were not included in the original review:

Line 46-47: "'Joerg Peninsula' and the approximate distance in km of the site from the MODIS Mosaic of Antarctica (MOA) grounding zone position"  this takes a second to understand and could be clarified by writing "a portmanteau of the location, 'Jeorg Penisula', and 21, the approximate distance in km..."

Line 56: clarify in which studies they were previously known as CI-O and WI-0 respectively

Line 98: "the downward logs were indistinguishable": clarify that this means indistinguishable from those taken moving up the borehole (otherwise it sounds like indistinguishable from the other borehole)

Line 108-109: I think you mean to say here that the strike is plotted as the angle, and the dip is the distance, relative to the centre, but this is not clear.

Line 263: unclear what 'spatially generalised' means

Reviewer #2

(Remarks to the Author)

Reviewer #3

(Remarks to the Author)

Reviewer #4

(Remarks to the Author)

I attach my review as a pdf, as it contains a figure.

Version 2:

Reviewer comments:

Reviewer #1

(Remarks to the Author)

I am satisfied by the edits made by the authors, and I am happy to recommend publication of this paper. Congratulations to the authors on a nice piece of science!

Reviewer #2

(Remarks to the Author)

Reviewer #3

(Remarks to the Author)

Reviewer #4

(Remarks to the Author)

Thanks for addressing my concerns, I do like the new figures in the supplement, and the additional hint in the figure caption that the sketch for the Solberg Inlet unit does not contain the 180 degree-rotated inclined layers.

I do not have any further points.

We thank all the reviewers for their positive and constructive reviews. We have responded point-by-point below, in blue italic font.

Review 1

Review of “Influence of grounding zone processes on the internal structure of ice shelves” by Miles et al. In this paper, the authors report the discovery of a new unit of ice located in ice shelves in Antarctica. This paper presents a study of the Larsen C ice shelf, combining data from optical televiewer images from four different boreholes alongside GPR interpretation and modelling of flow lines. The authors conclude that there is justification for a new intermediate layer between meteoric ice and continental meteoric ice based on the dip of these layers. Overall, the paper is well written, concise, and relatively easy to follow. The analysis carried out by the authors is very thorough, the findings notable and the discussion interesting. However, we find several aspects of the paper to be lacking and outline these below. We feel that the paper requires these comments to be addressed before publication in order for the reader to be able to validate the findings of the authors, and justify the significance of the research. We look forward to seeing this paper published (whether in Nature Communications or elsewhere), with the comments addressed, as we believe this will make the paper more balanced and impactful.

Thank you for the supportive review and helpful suggestions, which we have responded to individually below.

Major comments:

- Modelling and evidence for the mechanism: the modelling work provides only weak evidence to support the proposed mechanism. This is particularly important because it is the main piece of evidence presented to support the authors’ hypothesis of the mechanism of formation. At JP-21, the central estimate intersects well with the proposed location of the unit, but the bounds on the estimation are very large. On SI-47, the actual location of unit 2 is some distance away from the ‘best’ location. The plausible range occupies more than half of the ice shelf thickness, and it is therefore quite likely that the location of the proposed unit will intersect the band somewhere (even, for example, if the band has a randomly chosen location). A more robust test would have been to have run these simulations backwards, looking at the range of origins of flowlines which necessarily pass through the band. We suspect that, because of the large spread seen here, this could correspond to a very large range in origins, for example many 10s of kilometers around the grounding zone. Without this evidence, the proposed mechanism seems somewhat speculative, and the connection to grounding zone processes tenuous. The authors claim that they “cannot envisage any process of deformation capable of creating an ice layer with the properties of Unit 2”, but do not provide any evidence (nor references to support this claim). The proposed formation of steep topography at the grounding zone, could it be checked by existing data? This would greatly support the concept. (Aside: the title is potentially misleading: even if the proposed mechanism is to be believed, it really doesn’t have anything to do with grounding zone processes, but simply that there is a change in slope of the surface in the grounding zone, which is a consequence of the global behaviour of an ice sheet.)

Thank you for these suggestions. We have now altered the title to refer to “the grounding zone” rather than “grounding zone processes”. We have also conducted the flowline modelling in reverse for both the uppermost and lowermost boundaries of Unit 2 in both boreholes, the results of which we include below. Whilst this was a worthwhile endeavour to identify the source location of the top and base of Unit 2, which remains around the grounding zone, we prefer not to replace the original modelling, for two reasons. The first is that the flowline modelling is only a secondary means of confirming that Unit 2 could have been sourced around the grounding zone – our main evidence is based on the structure and characteristics of the optical televiewer-derived layers,

which vary with depth through each borehole. The second is that, despite updating the flowlines to each borehole, we were unable to begin the flowline modelling for JP-21 high enough that we could use the same uncertainty bounds for both sites. Figure 1 below (JP-21) therefore shows upper uncertainty ranges for the 10th percentile surface mass balance for the top of Unit 2 and the 25th percentile surface mass balance value for the base of Unit 2, instead of the minimum surface mass balance. Figure 2 (SI-47) shows the minimum surface mass balance value for the upper uncertainty range.

Figure 1 – Reverse flowline modelling results for JP-21. Note that the semi-transparent error bars represent the 10th percentile surface mass balance for the top of Unit 2 and the 25th percentile surface mass balance for the base of Unit 2.

Figure 2 – Reverse flowline modelling results for SI-47. The semi-transparent error bars represent the same surface mass balance uncertainty ranges as the initial model runs (inputs of min and max surface mass balance).

Finally, we would like to stress that this flowline modelling is not the primary evidence we present to support our hypothesis of the mechanism of formation. In fact, it has little bearing on the **mechanism** of formation (as noted above; that inference is based largely on the structure of the units concerned) – but it does have a clear bearing on the **location** of formation. The main evidence for the location of Unit 2 formation is that, across this and other ice shelves, there must be a material unit at depth that was accumulated upflow of the grounding zone, and there must be a shallower material unit, extending to the surface, accumulated downflow of the grounding zone (i.e., on the ice shelf). Given the location and properties of Unit 1, allied to the ice shelf surface being close to horizontal, it is beyond reasonable doubt that our Unit 1 accumulated on the ice shelf. Similarly, given the deeper location and contrasting structure of our Unit 4, it seems almost as certain that Unit 4 accumulated upflow of the grounding zone, above bedrock which is typically characterised by more complex englacial strain. If we accept that Unit 1 forms on the ice shelf and Unit 4 forms upflow of the grounding zone, it follows that our Unit 2 (and 3) form between the two - somewhere around the grounding zone. Irrespective of the details of the unit's process of formation (which we are not certain of but do propose a reasonable mechanism for), the flowline modelling was undertaken solely as an independent means of determining whether the depth of Unit 2 was **consistent with** its formation at the grounding zone. The analysis, whether run forward to the borehole or backwards from it, indicates clearly that it is consistent. In reality, the analysis is indeed subject to substantial uncertainty – and that is real and justified being based on historic velocities and accumulation rates – but we reiterate that the modelling represents an independent check on what we believe is a strong and reasonable deduction; it is not the primary line of evidence. We have adjusted the text, where relevant, to make this clearer.

- Radar coherency. The explanation provided in the paper does not adequately justify the claim that the lack of coherence in the GPR data indicates a new layer in the ice. Including a second example of this characteristic in the Larsen Ice Shelf would strengthen the argument, along with better annotation of the existing GPR profile. Is it possible to determine the strike and dip from the radargram? In the Brunt Ice Shelf data that is referenced (reference 9), the absence of layers in the meteoric ice is not attributed to an incoherent signal, but rather to the very old, compressed ice, which likely has layers that do not produce internal reflectivity. The statement, "Given the unit's likely ubiquity, this has implications for the interpretation of ice-penetrating radargrams from all ice shelves," is difficult to support with the radar data and details currently provided in the manuscript. Rather, it feels this layer in radargrams is already a characteristic of glaciers flowing into ice shelves. In addition, it is not justified that the incoherent features in the GPR are related to the proposed new unit of ice. They could also be related to ice fabric, steepness not detected by the acquisition strategy, or even homogeneous ice without layers which are so homogeneous that there is no strong englacial reflection.

Thank you for raising this issue. While we have no other profiles that intersect boreholes of sufficient depth, we have added a reference to Kulesa et al. (2019), which presented radargrams for elsewhere on Larsen C that are also characterised by a loss in radar energy return at depth. We agree that there are many other factors that may contribute to a loss of radar energy return at depth and we are here adding one more process to this list. We also acknowledge that the effect of increasingly steep internal layering on radar return is relatively unexplored theoretically (despite it being spatially systematic and extensive if our interpretation is correct). We have therefore rephrased the text to focus more carefully on the coincidence of the steeply dipping layers with the loss in radar energy return and the top of Unit 2. We have also added a sentence to the Discussion identifying some of the other factors that could be responsible (anisotropy, destructive interference in trace stacking, or off-nadir ray path losses), with references to the literature. We have additionally changed all instances of referring to radar coherence to loss in radar energy return, as this more accurately describes our observations.

We also provide more detail on the survey mode used in the main text and in the Methods, i.e., bistatic dipole antennas mounted at a distance of 1.4 m (as limited by the dimensions of the plastic sledge used) in a perpendicular-broadside configuration. Here, we now go on to recommend the use of potentially better-suited modes of data acquisition in such settings, such as, e.g., bistatic antennas mounted in a parallel-broadside configuration (as considered e.g. by Langhammer et al. (2017) in helicopter radar surveys on Alpine glaciers, now cited in the text) or wide-angle radar reflection (WARR).

- Definition of a new unit. We struggle to support the justification that the variation in dip of these layers is criteria for a new unit. These layers are for all intents and purposes the same characteristics as the meteoric in situ ice in unit 1 (as understood from the paper). The process and location of formation – if correctly justified – is interesting and worthy of publishing, but the description of it being a new type of unit does not seem justified. If this section were supported by physical properties of this dipping unit being different to the unit above, then this would help support the cause for a new definition of the unit. Or if the location of this layer were supported by more thorough modelling then its presence would be more significant and justification for noting its location. *The units were defined on the basis of their physical characteristics before they were interpreted. These characteristics included notably different trends in dip (which is < 20° for Unit 1 and increases from 20 - 80° for Unit 2) and strike (which has no strong directional trend for Unit 1, but does for Unit 2). We therefore believe there to be sufficient evidence to define Unit 2 as being structurally distinctive from Unit 1. Considered another way: had we left Unit 2 as part of Unit 1, we would need to make a distinction between the majority of Unit 1*

and “the lowermost section of Unit 1 where layers rotate and orientate”. We believe that needing to make (repeatedly) such a within-unit distinction could be confusing and unhelpful in terms of interpretation.

- Paragraph from lines 242 to 253 is very interesting because it brings to debate several important questions. However, there are no answers or explored possibilities, and surely they could enrich the discussion.

We have included some explored possibilities in this paragraph where possible, for example, for points three (the orientation influence of lower units on Unit 1, which we assume weakens as further ice accumulates and layers shallow, perhaps partly due to snow redistribution on the surface) and four (not considering flexure as the ice becomes buoyant, which we assume will not have a large effect but have no way of evaluating this claim).

- Figures: All figures were hard to interpret and follow without the names of the boreholes listed on them, and would benefit from additional annotation, and explanations of what the X and Y axis are showing, especially for the optical televiewer data. Figure 1: Blue and red lines around the square boxes in led to confusion as the colours are very similar to those used to represent the flow lines and radar profile. A change of these colours would make it easier to follow.

We have revised all figures to make them easier to interpret. Figures 2, 4, 5, 6, and 7 now include the borehole names on the relevant panels. We have changed the outline colour for the panels in Figure 1 and the background image (as requested in the other review). Figures 3, 4, and 8 now explain in the caption that strike is relative to true North, and the Figure 4 caption also indicates how to read strike and dip from the stereoplots. We now also include two additional supplementary figures: one demonstrating the inclination of each borehole with depth; and one demonstrating layers that were delineated from a deep section of log from each borehole.

Minor Comments:

- The concepts “strike” and “dip” should be described for readers not experienced on Kamb contours. This description could be supported by strike and dip marks in Figure 2B-D and Figure 2-F-H. Figure 3 A,C could include markers at strikes 0°, 90°, 180°, 270°, and also in Figure 4, for better traceability by the reader. Is the strike angle 0° oriented to the North? Maybe an orientation relative to the flow line or to other features could be more supportive for the reader.

In the caption for Figure 4, we have now included both a reference for Kamb contours and explained how strike and dip can be read from the hemispheric plots (we prefer not to include this information on the figure as it would become crowded and obscure some data points). We now include a supplementary figure that shows the strike and dip of delineated layers for deep sections of both boreholes, rather than include this information on Figure 2, which would somewhat obscure the layers (which we now indicate with red arrows). We have also revised Figure 3 so that: i) the panels include the borehole names; ii) the strike markers are at 0/90/180/270/360° and dip markers at 0/30/60/90°; and iii) the coloured points are larger to aid visibility. In the captions for Figures 3, 4, and 8, we have added a sentence explaining that strike is relative to true North, which is also now mentioned in the Methods.

- The sub-figures could guide the reader better if they were clearer referenced in the text. For example, in lines 77-78, referencing several of these sub-figures at once with “Figure 2B, C, F, G”, and in line 162 with “Figures 2D and H”, could be confusing. The depth axes do not always correspond to the text description, like for example: in line 77 with “103-134 m”; in lines 113-114 the “depth of 83 m” does not correspond to the “range: 45 -124

m”; in line 114 the “age of 139 years” does not fit well with the time in Figure 6A; in line 114 the “depth of 161 m” compared to “range: 88 -237 m”; and in line 146 the thicknesses of Unit 2 could be marked in Figure 8.

In the revised manuscript, we have ensured that all figure panels are more precisely referred to in the text and ensured that depth ranges in the text are correct. On Figure 8, we already mark (with a red dashed line) the interpretation of the original analysis of the boundary between in situ meteoric ice (akin to our Unit 1) and continental meteoric ice (akin to our Unit 4). We prefer not to re-interpret our units onto the figure itself, particularly as borehole WI-28 is not deep enough to show all of Unit 2.

- Figure 6: Instead of the median of the evolved grounding zone depth, would the mean be valid?

We are not referring to the median grounding zone depth here, but the median surface mass balance input to the model run. We believe the mean would be valid and have tested the variation between these two inputs, and found the resulting depth difference to be minor compared to our other estimated uncertainties, so have opted not to change from median to mean.

- Figure 1. Line 158: What does the borehole water layer have to do with unit 1?

The reviewer is correct in questioning the relevance of the water level to interpretations and conclusions of this paper. However, it is relevant to borehole logging, whereby instruments are more stable in terms of high-frequency movement in liquid-filled boreholes than they are in air-filled boreholes. More generally, the borehole water limit indicates the pore close-off depth, which is useful for people interested in densification and firn saturation, and may therefore be useful for comparison.

- Figure 9: It is a very illustrative figure and it really helps the reader; in addition, could Unit 3 be included?

Thank you – we are grateful to know that Figure 9 is helpful. We prefer not to include Unit 3 in the figure as it would complicate the systematic pattern we illustrate with what was in effect an ad hoc occurrence. However, we take the reviewer’s point and now include a statement in the caption specifying where Unit 3 is located in the JP-21 borehole.

- Line 242: We do not understand the explanation “despite their contrasting flowlines”, because both flow line orientations seem similar. What do you mean by “contrasting”?

Here, we are referring to the different flowline directions at the grounding zone. We have now changed this text to: “despite their contrasting flowline directions at the grounding zone”.

- Line 287: Instead of V, would it be W (watts)?

Thanks for checking, but we do mean V here when referring to the transmitter power.

- Data availability: in the UK PDC link provided for the “logs from northern Larsen C”, the section “PROJECT HOME PAGE” has a link to not the intended page.

Thanks for letting us know – we have requested the links be removed from the UK PDC pages. Whilst the project web pages are no longer maintained, the project’s datasets are still available from the PDC.

We would like to conclude by saying that once again, we believe this paper presents excellent science and thorough research, and the major comments should not be taken to be disheartening, but rather they should allow the paper to provide greater impact with a well-rounded discussion.

Review 2

In this manuscript borehole televiewer logs from the Larsen C ice shelf are evaluated, from hot water drills close to the grounding zone in the southern part of the ice shelf. The data reveal a change in the dip of internal layers, which is then interpreted as an inherited feature from the transition at the grounding zone. I think this could be an interesting study, but I have some major concerns, which I hope can be addressed by the authors.

Thank you for the supportive review and helpful suggestions, which we have responded to individually below.

1. I do not understand the definition of the Units. This definition seems to be based on Figure 3. I tried to understand the apparent two-mode status (with a 180 degree difference, which made me suspicious) of the strike direction. This became then clear by looking at Figure 4, the stereoplots for the planes of the layers. From here it is clear that the poles to planes of Unit 2 of SI-47 are undulating around zero, which explains the change in strike by 180 degrees. I think the stereoplots give a much better illustration of the geometry, and are also more suitable to define Units. Here it looks like the dip of planes of the layers in SI-47 are undulating back and forth, until finally stagnating towards one side with a high dip angle. For JP-21 this stage is missing, and the dip is just increasing with depth in the same direction (except for the funny Unit 3). I think this should be taken into account when interpreting the data, and it might change conclusions, as in that case there would be no gradual change of dip in one direction, as shown in fig. 9. What I did not get from the manuscript: Is the strike direction relative to geographic north? How does this then relate to the flow direction?

We defined our units based on the physical characteristics of the layers, including the differing trends in dip (which is $< 20^\circ$ for Unit 1 and increases from $20 - 80^\circ$ for Unit 2) and strike (which has no strong directional trend for Unit 1, but does for Unit 2). The definition primarily used the optical televiewer-derived structural characteristics, including layer type, scale, frequency, strike, and dip – the latter two being demonstrated in Figure 3. While we agree that the stereoplots are useful, we could not use them on their own to define the units because they do not demonstrate the variability in layer structure with depth (e.g., such as is central to the definition of Unit 2). Very few layers in Unit 2 are near 0° , so the undulation around zero on the stereoplots does not explain the 180° change in strike (this is something we mention in the Discussion that we cannot explain, but do know that it does not result from instrumental error). The optical televiewer image logs are measured relative to magnetic North and we rotate the logs, and therefore strike, to true North – we have now included mention of this in the Methods section, the main text, and relevant figure captions.

2. I think there is some essential Information missing. Is the bore hole shape logged? There is no information on how the borehole diameter changes, how the inclination of the borehole changes. This might well have an impact on the appearance of the layers in the images.

Thank you for checking this. Whilst we did not log the borehole cross-section, it was recorded in JP-21 by directional video. This indicated substantial variability in the borehole cross-section within the uppermost ~ 30 m (a result of drilling progressing through alternating layers of permeable snow/firn and impermeable ice) but no significant deviation in shape below this depth. Borehole inclination was recorded by both directional video and optical televiewer, and typically varied by less than a degree from vertical. We now note this in the Results and include a plot of inclination against depth in the supplementary information.

3. Is the image log somehow orientated? Can the logger rotate in the hole, or is this somehow corrected?

Yes, the optical televiewer logs are automatically orientated to magnetic North (by on-board magnetometers), so probe rotation in the borehole is corrected for. We then rotated the image logs to be relative to true North. We now state in the Methods and all relevant figure captions that image logs and strike values are relative to true North.

4. The interpretation that the inclined layers are linked to the slope of the glaciers before entering the ice shelf is interesting, but I think it would be good to have a look at the actual slopes and show some data here. 60 degrees seems quite a lot. Why is the “final” dip angle so similar in both boreholes, when one is from a rotated block in the suture zone, and one from a smooth transition?

We have investigated the surface and basal slopes at both sites from BedMachine3, which shows slopes of $\sim 20^\circ$ at SI-47. While this is substantially lower than the dip of the layers at the base of Unit 2 and through Unit 4 (up to $\sim 80^\circ$), BedMachine has a spatial resolution of 0.5 km and is subject to significant uncertainty. In reality, therefore, local slopes will be substantially higher than 20° – but we do not know by how much. Nonetheless, the difference between the 20° given by satellite and the $\leq \sim 80^\circ$ we record is substantial and – in the absence of measurements of the englacial structure of the feeder glaciers upflow of the grounding zone – we are not aware of any other specific process to invoke. This is why we are careful in our language to go no further than to suggest possible mechanisms to attain such steep slopes. We believe it is a coincidence that the dip angle of the deepest layers is similar in both boreholes, though it is perhaps possible that the suture zone grounded ice did not rotate when joining the floating ice shelf.

5. I think in the transition from inclined to horizontal layers in the transition zone (shown in figure 9) should be discussed a little more. Is this only geometry, or locally variable accumulation (why should it be different?)

We envisage the shallowing of layers through Unit 2 occurs primarily as a result of the shallowing of the ice shelf surface through and beyond the grounding zone. However, it is certainly possible that systematic redistribution of accumulation may also play a role, and we have added mention of this into the Discussion.

comments on the figures:

- in figure 3, 4, 6 and 8, it would be really nice to have the names of the boreholes in the panels, instead just small in the figure caption.

We have now added borehole names onto the panels in Figures 3, 4, 6, and 8.

- In figure three it is difficult to see the few colored dots in the more crowded parts. Maybe make them also bigger to stand out?

We have now changed the coloured data points in Figure 3 so that they are larger and easier to identify.

- For figure 2 it would be good to add a simple sketch of the geometry, for example the cylindrical borehole with an oblique plane intersecting it, and then “unroll” it to show the resulting curve. It would be also good to show in the zoom-panels which curves are picked from these. For me it is also not that obvious what is crevasse and what is not in panel D and H, so maybe annotate or add an arrow. And maybe stick to the same depth scale as the other panels.

In the Methods section, we have referenced Hubbard et al. (2008), which presents such a figure showing a cylindrical borehole with an oblique plane intersecting it and the unrolled curve; to save space, we choose not to

repeat it here. In panels D and H, we have added coloured arrows to annotate the crevasse traces. We prefer not to change the depth scale of these panels as that would result in the loss of layer detail. We have now included a supplementary figure that shows example layer delineations for a section of log deep in each borehole (panels D and H from Figure 2, showing crevasse traces).

- When looking at the lower part of the logs in figure 2 the dip-angles should be similar (as shown in fig 3), but JP-21 looks considerably less wavy (above the black part of course). Is this just an illusion? Maybe it would be good to add a supplementary figure with a comparison, with a scale between the overview image and the zoom panels.

We believe this is an illusion that is perhaps more evident in the normalised greyscale logs than in the true-colour logs. We now include a supplementary figure that shows example layer delineations, including for the deepest section of log shown in panel D in Figure 2. The example layer delineations, particularly for the crevasse traces, demonstrate the high-angle layers that are present in this part of the log.

- Figure 1: The background optical image does not show a lot of structure of the ice shelf surface. Maybe it would be better to use a radar image, where the flow units and suture zone can be better distinguished.

We investigated some Sentinel-1 SAR images, but did not feel that these showed the surface structure any more clearly, so we have instead replaced the background image with a MODIS scene from 2016.

We thank all the reviewers for their positive and constructive reviews. We have responded point-by-point below, in blue italic font.

Review 1

After the revision, the authors address most of the comments of the original submission. However, some of the additions were not clearly indicated in their response letter. Also, we think that some of the responses could be included in the final version, for in case other readers have the same questions.

Thank you for the supportive review and helpful suggestions. We respond to the comments below and try to clearly indicate all the additions made. We also include the requested responses from the last review in the revised manuscript.

Regarding the major comments: All were nicely addressed, except the next two:

1. Modelling and evidence for the mechanism...

Figures 1 and 2 of the response could be included as supplementary figures. They are also very illustrative and worth including if the authors do not want to replace the original document, and would support the authors' suggested location of formation, as we outlined in our initial review. If the authors want to include them, are the percentiles 10th and 25th relative to the maximum or to the minimum accumulation rates?

We now include these figures as a supplementary figure and refer to them in the revised manuscript where appropriate. We also describe the ranges more clearly in the caption: "The uncertainty bounds are shown by light shaded regions: red for the upper boundary of Unit 2 are the 10–90th percentiles for JP-21 and minimum–maximum surface mass balance values for SI-47; blue for the lower boundary of Unit 2 are the 25–75th percentiles for JP-21 and minimum–maximum surface mass balance values for SI-47. These vary because the flowline modelling for JP-21 could not start high enough that the same uncertainty bounds (min-max as used in the Figure 6) could be used for both sites." We have also added information about these later model runs to the Methods section.

In the response, authors say 'The analysis, whether run forward to the borehole or backwards from it, indicates clearly that it is consistent. In reality, the analysis is indeed subject to substantial uncertainty', and we think this is important to be mentioned in the article. Maybe could it be included in the paragraph between lines 180-189 (where the additions on the formation of Unit 2)?

We have included this in the revised manuscript.

The new lines 321-323 for further detailing the survey greatly help the reader. We would rephrase with 'In the bistatic configuration, the dipole antennas were mounted... ', or similar, as the antennas are not bistatic, but the system.

We have made this change in the revised manuscript both here and in the main text.

2. Radar coherency

The response and the revised text address our concerns, except when we asked about the possibility of determining the strike and dip from the radargram. In lines 140-141 and 234-249 the authors say that Units 2 and 4 cannot be identified in the radargram of Figure 7, but maybe within Unit 1 can the strike and dip be estimated?

It would be possible to estimate the strike and dip of Unit 1 from the radargrams, but this will be much less accurate than calculating from the layers in the optical televiewer image logs, so in our opinion, there is no additional benefit to such an estimation. We had in fact acquired radargrams at additional azimuths to aid visualisation of internal ice layering, but due to a major malfunction of the radar control unit (which was eventually repaired by the manufacturer) the majority of the data were unfortunately lost. We will consider

this again in future work, but for now, we feel that the optical televiewer-derived layer strikes and dips fully support the main conclusions of the manuscript regarding layer dips and strikes.

Regarding the minor comments: All were well addressed, except the next two:

1. The sub-figures could guide...

Except in Figure 8, we do not see some of the changes we recommended, but if the authors think this way is better, we accept it.

We did revise Figure 8 by adding borehole names onto the panels and adding an explanation of strike into the caption. We opted not to put the additional interpretations on Figure 8 for the reasons specified in the previous response: "On Figure 8, we already mark (with a red dashed line) the interpretation of the original analysis of the boundary between in situ meteoric ice (akin to our Unit 1) and continental meteoric ice (akin to our Unit 4). We prefer not to re-interpret our units onto the figure itself, particularly as borehole WI-28 is not deep enough to show all of Unit 2."

2. Line 287: Instead of V, would it be W (watts)?

We might be misinterpreting what the authors say, but power units are watts, not volts as seems from V units. Although this is not related to the quality of the article, we think it should be corrected.

The output of the Sensors & Software Transmitters is given in V(olts), not W(atts), please see: <https://qef.nerc.ac.uk/equipment/pepro/>. Two models of transmitter are available, 400 V or 1000 V, and we used the 1000 V transmitter. We have rephrased this to: "a high-power transmitter (transmitter output: 1000 V)", which we hope clarifies this concern.

A few more minor comments which were not included in the original review:

Line 46-47: "'Joerg Peninsula' and the approximate distance in km of the site from the MODIS Mosaic of Antarctica (MOA) grounding zone position"  this takes a second to understand and could be clarified by writing "a portmanteau of the location, 'Jeorg Peninsula', and 21, the approximate distance in km..."

We have revised this description to: "an acronym of the location (Joerg Peninsula) and the approximate distance in km (21) of the site from..."

Line 56: clarify in which studies they were previously known as CI-O and WI-0 respectively

We have added references to the relevant studies at the end of this clause.

Line 98: "the downward logs were indistinguishable": clarify that this means indistinguishable from those taken moving up the borehole (otherwise it sounds like indistinguishable from the other borehole)

We have changed this to: "the downward log was indistinguishable from the upward log in each borehole".

Line 108-109: I think you mean to say here that the strike is plotted as the angle, and the dip is the distance, relative to the centre, but this is not clear.

We have changed this to: "The strike ... is plotted as the angle, and the dip as the distance from the centre ... of the plots".

Line 263: unclear what 'spatially generalised' means

We have changed this sentence to: "In the absence of further evidence, we consider this to be a geometrical coincidence."

Review 2

I am reviewing this manuscript for the second time, and most of my comments and suggestions have been addressed. But my main concern is still there, and I'll try to make it more clear what I mean by this. In my first report I wrote:

1. I do not understand the definition of the Units. This definition seems to be based on Figure 3. I tried to understand the apparent two-mode status (with a 180 degree difference, which made me suspicious) of the strike direction. This became then clear by looking at Figure 4, the stereoplots for the planes of the layers. From here it is clear that the poles to planes of Unit 2 of SI-47 are undulating around zero, which explains the change in strike by 180 degrees. I think the stereoplots give a much better illustration of the geometry, and are also more suitable to define Units. Here it looks like the dip of planes of the layers in SI-47 are undulating back and forth, until finally stagnating towards one side with a high dip angle.

In the reply the authors state that there are only a few points close to zero. But this is not the point, it is the undulation around zero. In my interpretation of the stereo plots in figure 4 in combination with the information in figure 3, the layers in Unit 2 from the Solberg inlet are changing strike by tilting back and forth with depth, and not changing its dip angle consistently into the same direction. This would look like this approximately, two strike directions 180 degrees apart, more smooth changes in dip angle:

This also would mean that the panel on the left side of figure 5 is misleading, as it is only showing a change in dip angle with depth, and not the 180° changes of the strike. It is mentioned in the manuscript that the Unit 2 has two major strike directions, but it should be discussed that it is a 180° change.

This would also mean that the conceptual figure 9 does not fit to the shown data, at least for the Solberg Inlet. This is not an unusual pattern and can be seen in ground penetrating radar data which has been gathered for information about accumulation on the ice sheets for example, where it is thought to be linked to accumulation and advection, maybe not with these high dip angles though.

Thus, if I am interpreting the data from the plots in the right way, the authors would have to reconsider their model for explaining the observed pattern and adapt the discussion.

We agree with the reviewer on this point and note that the revised manuscript did address the presence of contrastingly orientated layers within Unit 2 at SI47. For example, lines 183-184 noted: "These dips tend to ... two dominant orientations at SI-47 (Figures 3 and 4)" and lines 258-261 noted: "We note that our unit formation interpretations leave several unresolved aspects... First, why Unit 2 at SI-47 has two dominant orientations, 180° out-of-phase – the presence of the same layers (with the same orientation) in both the downwards and upwards logs excludes instrumental error". Nonetheless, we thank the reviewer for making it clear that this complication should have been addressed more explicitly. The revised manuscript now does this by including the following:

- We have emphasised that the two dominant orientations of Unit 2 at SI-47 are 180° different in strike in the following locations in the tracked-change revised manuscript:
 - i. Line 85: “...strike tends to one (at JP-21) or two (at SI-47; differing by ~180°) orientations...”
 - ii. Lines 93-97: “The primary eigenvalues indicate a single preferred orientation... except for Unit 2 at SI-47, which has a second preferred orientation that is 177.1° different in strike from the orientation of the primary eigenvector. For this unit at SI-47, the primary eigenvalue is 0.76 and its secondary eigenvalue is 0.24”
 - iii. Lines 123-125 (added to the Figure 5 caption): “While most Unit 2 layers conform to the orientations shown in the expanded panel for SI-47, several are oriented at ~180° to this primary orientation while maintaining a similarly increasing dip with depth (see Figure 3C and D and Supplementary Figure 3).”
 - iv. Lines 197-199: “These dips tend to ... two dominant orientations (~180° different in strike) at SI-47 (Figures 3 and 4, Supplementary Figure 3).”
 - v. Lines 208-209: “...comprised of layers that either have a single, dominant orientation or have two dominant orientations that are ~180° different...”
- We have included an additional supplementary figure, copied in below, that shows Unit 2 layers as a tadpole plot. We now refer to this figure in the main text and in the caption for Figure 5. We prefer to keep it in the supplement because it presents the same information as Figure 3 in a different way.

In addition, in the revised manuscript, we now supplement our existing suggested process of formation for Unit 2 – i.e., that based on change of slope across the grounding zone – with longitudinally compressive deformation as ice exits the grounding zone. The revised Interpretation section (lines 205-242 in the tracked-change revised manuscript) now reads as follows:

“Any process advanced for the formation of Unit 2 needs to explain as many of its distinctive properties as possible. These properties are that it is: i) located at the top of the ice column at the grounding zone; ii) tens of metres thick; iii) comprised of layers that shallow, from ~60° to ~0°, upwards, and are hence conformable with Unit 1 layering above and Unit 4 layering below; and iv) comprised of layers that either have a single, dominant orientation or have two dominant orientations that are ~180° different. While we know of no documented processes that can explain all of these properties, we hypothesise that Unit 2 forms through a combination of accumulation across a shallowing surface slope and intense deformation, both resulting from the substantial change in slope as ice flows from its steep grounded feeder glaciers into the approximately horizontal ice shelf (Supplementary Figure 5). Key elements with the capacity to create variable accumulation across a shallowing surface are outlined schematically in Figure 9. Here, in the grounding zone, Unit 4 is tilted to dip steeply while flowing over the steep bedrock below, and thus the first layers to accumulate on top of this unit maintain a similar dip and strike. Over time and with distance over the grounding zone, the surface slope of the ice shelf shallows, and subsequent accumulating layers also decrease in dip until the surface is near-horizontal. Such a process would result in a generally progressive increase in dip with depth – characteristic of Unit 2 at all sites where it has been logged by optical televiewer (i.e., JP-21 and SI-47 in the south and CI-27 and WI-28 in the north of Larsen C Ice Shelf; Figure 1). However, this process alone does not explain the significant minority of layers in Unit 2 at SI-47 with a strike rotated 180° from that of the dominant layering. These secondary layers are challenging to explain, particularly since they appear at SI-47 and not at JP-21. In the absence of additional evidence, we propose tentatively that these secondary layers form as fractures opening orthogonal to the main layering at locations where longitudinal compression is sufficiently intense to buckle and fracture the surface layers of ice as it flows into the shelf. In general, ice shelf strain is high across the grounding zone^{4,7,26,27} and particularly high along the edges of promontories where steep and fast-moving subsidiary glaciers emerge into the main trunk (Supplementary Figure 6). Formation of Unit 2 in

such a setting is consistent with the reconstructed flowline of the ice intercepted by SI-47 (Figure 1C), which indicates formation along the lateral margin of Solberg Inlet. In contrast, the ice intercepted by JP-21 crosses the grounding zone at the tip of Joerg Peninsula, a zone of more subdued longitudinal compression.”

We have also revised the caption for Figure 9 to:

“Conceptual illustration of the proposed mechanism to create variable accumulation across a shallowing surface unit formation around the grounding zone of an ice shelf”

And revised the wording in the Abstract to:

“We hypothesise that this unit forms as a result of changes in the surface slope of feeder glaciers as they flow over the grounding zone, resulting in both variable surface accumulation and intense deformation.”

Supplementary Figure 3: Tadpole plots for all delineated layers in Unit 2 for each borehole. The dip of each layer is indicated by the grey circles, with values read off the x-axis. The strike of each layer is plotted by the tail of each circle as the angle relative to Antarctic Stereographic Polar North, so as to be directly comparable to Figure 5 in the main manuscript.

Second, there are much better options than BedMachine for a high-resolution DEM, for example this here: <https://nsidc.org/data/nsidc-0516/versions/1>

This should give at least a better clue on how likely it is to find 60° slopes around the peninsula.

I hope these issues can be resolved or adapted, as I find this is an interesting approach and a good idea to combine the televiewer and radar data to learn about the history of the ice.

We thank the reviewer for this suggestion. We have calculated slope from the 8 m resolution TanDEM-X DEM that we use elsewhere in the manuscript (cf. 100 m for the ASTER DEM and 500 m for BedMachine) – shown

below. We note that, as expected and noted in the manuscript, there are indeed steep slopes, including some $> 60^\circ$ along the Joerg Peninsula. We have added this figure to the revised manuscript as Supplementary Figure 5.

Review Figure 1: Surface slope over the Joerg Peninsula, calculated from the TanDEM-X DEM.

Review of “Influence of grounding zone processes on the internal structure of ice shelves” by Miles et al.

In this paper, the authors report the discovery of a new unit of ice located in ice shelves in Antarctica. This paper presents a study of the Larsen C ice shelf, combining data from optical televiewer images from four different boreholes alongside GPR interpretation and modelling of flow lines. The authors conclude that there is justification for a new intermediate layer between meteoric ice and continental meteoric ice based on the dip of these layers.

Overall, the paper is well written, concise, and relatively easy to follow. The analysis carried out by the authors is very thorough, the findings notable and the discussion interesting. However, we find several aspects of the paper to be lacking and outline these below. We feel that the paper requires these comments to be addressed before publication in order for the reader to be able to validate the findings of the authors, and justify the significance of the research. We look forward to seeing this paper published (whether in Nature Communications or elsewhere), with the comments addressed, as we believe this will make the paper more balanced and impactful.

Major comments:

- Modelling and evidence for the mechanism: the modelling work provides only weak evidence to support the proposed mechanism. This is particularly important because it is the main piece of evidence presented to support the authors' hypothesis of the mechanism of formation. At JP-21, the central estimate intersects well with the proposed location of the unit, but the bounds on the estimation are very large. On SI-47, the actual location of unit 2 is some distance away from the 'best' location. The plausible range occupies more than half of the ice shelf thickness, and it is therefore quite likely that the location of the proposed unit will intersect the band *somewhere* (even, for example, if the band has a randomly chosen location).

A more robust test would have been to have run these simulations backwards, looking at the range of origins of flowlines which necessarily pass through the band. We suspect that, because of the large spread seen here, this could correspond to a very large range in origins, for example many 10s of kilometers around the grounding zone.

Without this evidence, the proposed mechanism seems somewhat speculative, and the connection to grounding zone processes tenuous. The authors claim that they “cannot envisage any process of deformation capable of creating an ice layer with the properties of Unit 2”, but do not provide any evidence (nor references to support this claim). The proposed formation of steep topography at the grounding zone, could it be checked by existing data? This would greatly support the concept.

(Aside: the title is potentially misleading: even if the proposed mechanism is to be believed, it really doesn't have anything to do with grounding zone *processes*, but simply that there is a change in slope of the surface in the grounding zone, which is a

consequence of the global behaviour of an ice sheet.)

- Radar coherency. The explanation provided in the paper does not adequately justify the claim that the lack of coherence in the GPR data indicates a new layer in the ice. Including a second example of this characteristic in the Larsen Ice Shelf would strengthen the argument, along with better annotation of the existing GPR profile. Is it possible to determine the strike and dip from the radargram? In the Brunt Ice Shelf data that is referenced (reference 9), the absence of layers in the meteoric ice is not attributed to an incoherent signal, but rather to the very old, compressed ice, which likely has layers that do not produce internal reflectivity. The statement, "Given the unit's likely ubiquity, this has implications for the interpretation of ice-penetrating radargrams from all ice shelves," is difficult to support with the radar data and details currently provided in the manuscript. Rather, it feels this layer in radargrams is already a characteristic of glaciers flowing into ice shelves.

In addition, it is not justified that the incoherent features in the GPR are related to the proposed new unit of ice. They could also be related to ice fabric, steepness not detected by the acquisition strategy, or even homogeneous ice without layers which are so homogeneous that there is no strong englacial reflection.

- Definition of a new unit. We struggle to support the justification that the variation in dip of these layers is criteria for a new unit. These layers are for all intents and purposes the same characteristics as the meteoric in situ ice in unit 1 (as understood from the paper). The process and location of formation – if correctly justified – is interesting and worthy of publishing, but the description of it being a new type of unit does not seem justified. If this section were supported by physical properties of this dipping unit being different to the unit above, then this would help support the cause for a new definition of the unit. Or if the location of this layer were supported by more thorough modelling then its presence would be more significant and justification for noting its location.
- Paragraph from lines 242 to 253 is very interesting because it brings to debate several important questions. However, there are no answers or explored possibilities, and surely they could enrich the discussion.
- Figures: All figures were hard to interpret and follow without the names of the boreholes listed on them, and would benefit from additional annotation, and explanations of what the X and Y axis are showing, especially for the optical televiewer data. Figure 1: Blue and red lines around the square boxes in led to confusion as the colours are very similar to those used to represent the flow lines and radar profile. A change of these colours would make it easier to follow.

Minor Comments:

- The concepts “strike” and “dip” should be described for readers not experienced on Kamb contours. This description could be supported by strike and dip marks in Figure 2B-D and Figure 2-F-H. Figure 3 A,C could include markers at strikes 0°, 90°, 180°, 270°, and also in Figure 4, for better traceability by the reader. Is the strike angle 0° oriented to the North? Maybe an orientation relative to the flow line or to other features could be more supportive for the reader.
- The sub-figures could guide the reader better if they were clearer referenced in the text. For example, in lines 77-78, referencing several of these sub-figures at once with “Figure 2B, C, F, G”, and in line 162 with “Figures 2D and H”, could be confusing. The depth axes do not always correspond to the text description, like for example: in line 77 with “103-134 m”; in lines 113-114 the “depth of 83 m” does not correspond to the “range: 45 -124 m”; in line 114 the “age of 139 years” does not fit well with the time in Figure 6A; in line 114 the “depth of 161 m” compared to “range: 88 -237 m”; and in line 146 the thicknesses of Unit 2 could be marked in Figure 8.
- Figure 6: Instead of the median of the evolved grounding zone depth, would the mean be valid?
- Figure 1. Line 158: What does the borehole water layer have to do with unit 1?
- Figure 9: It is a very illustrative figure and it really helps the reader; in addition, could Unit 3 be included?
- Line 242: We do not understand the explanation “despite their contrasting flowlines”, because both flow line orientations seem similar. What do you mean by “contrasting”?
- Line 287: Instead of V, would it be W (watts)?
- Data availability: in the UK PDC link provided for the “logs from northern Larsen C”, the section “PROJECT HOME PAGE” has a link to not the intended page.

We would like to conclude by saying that once again, we believe this paper presents excellent science and thorough research, and the major comments should not be taken to be disheartening, but rather they should allow the paper to provide greater impact with a well rounded discussion.

Report on Miles et al: "Influence of grounding zone processes on the internal structure of ice shelves"

I am reviewing this manuscript for the second time, and most of my comments and suggestions have been addressed. But my main concern is still there, and I'll try to make it more clear what I mean by this. In my first report I wrote:

1. I do not understand the definition of the Units. This definition seems to be based on Figure 3. I tried to understand the apparent two-mode status (with a 180 degree difference, which made me suspicious) of the strike direction. This became then clear by looking at Figure 4, the stereoplots for the planes of the layers. From here it is clear that the poles to planes of Unit 2 of SI-47 are undulating around zero, which explains the change in strike by 180 degrees. I think the stereoplots give a much better illustration of the geometry, and are also more suitable to define Units. Here it looks like the dip of planes of the layers in SI-47 are undulating back and forth, until finally stagnating towards one side with a high dip angle.

In the reply the authors state that there are only a few points close to zero. But this is not the point, it is the undulation around zero. In my interpretation of the stereo plots in figure 4 in combination with the information in figure 3, the layers in Unit 2 from the Solberg inlet are changing strike by tilting back and forth with depth, and not changing its dip angle consistently into the same direction. This would look like this approximately, two strike directions 180 degrees apart, more smooth changes in dip angle:

This also would mean that the panel on the left side of figure 5 is misleading, as it is only showing a change in dip angle with depth, and not the 180° changes of the strike. It is mentioned in the manuscript that the Unit 2 has two major strike directions, but it should be discussed that it is a 180° change.

This would also mean that the conceptual figure 9 does not fit to the shown data, at least for the Solberg Inlet. This is not an unusual pattern and can be seen in ground penetrating radar data which has been gathered for information about accumulation on the ice sheets for example, where it is thought to be linked to accumulation and advection, maybe not with these high dip angles though.

Thus, If I am interpreting the data from the plots in the right way, the authors would have to reconsider their model for explaining the observed pattern and adapt the discussion.

Second, there are much better options than BedMachine for a high-resolution DEM, for example this here: <https://nsidc.org/data/nsidc-0516/versions/1>
This should give at least a better clue on how likely it is to find 60° slopes around the peninsula.

I hope these issues can be resolved or adapted, as I find this is an interesting approach and a good idea to combine the televiewer and radar data to learn about the history of the ice.